# Drivers of Lassa fever in an endemic area of southwestern Nigeria (2017–2021): An epidemiological study

Simeon Cadmus[1,2,3,4]*, Victor Akinseye[2,4,5], Eniola Cadmus[6], Gboyega Famokun[7], Stephen Fagbemi[7], Gabriel Ogunde[8], Ayuba Philip[9], Rashid Ansumana[10], Adekunle Ayinmode[3,11], Olalekan Taiwo[12], Daniel Oluwayelu[3,13], Oyewale Tomori[14], Solomon Odemuyiwa[3,15]

1 Department of Veterinary Public Health and Preventive Medicine, University of Ibadan, Ibadan, Nigeria, 2 Damien Foundation Genomic and Mycobacteria Research and Training Centre, University of Ibadan, Ibadan, Nigeria, 3 Centre for Control and Prevention of Zoonoses, University of Ibadan, Ibadan, Nigeria, 4 Department of Microbiology, Nigerian Institute of Medical Research, Yaba, Lagos, Nigeria, 5 Department of Chemical Sciences, Augustine University, Ilara-Epe, Lagos, Nigeria, 6 Department of Community Medicine, College of Medicine, University of Ibadan, Ibadan, Nigeria, 7 Department of Epidemiology and Disease Control, Ondo State Ministry of Health, Akure, Ondo State, Nigeria, 8 Department of Epidemiology and Medical Statistics, College of Medicine, University of Ibadan, Ibadan, Nigeria, 9 Federal Ministry of Agriculture and Rural Development, Abuja, Nigeria, 10 School of Community Health Sciences, Njala University, Bo, Sierra Leone, 11 Department of Veterinary Parasitology and Entomology, University of Ibadan, Ibadan, Nigeria, 12 Department of Geography, University of Ibadan, Ibadan, Nigeria, 13 Department of Veterinary Microbiology, University of Ibadan, Ibadan, Nigeria, 14 African Centre of Excellence for Genomics of Infectious Diseases, Redeemer's University, Ede, Osun State, Nigeria, 15 Department of Veterinary Pathobiology, University of Missouri, Columbia, Missouri, United States of America

* simeonc5@gmail.com

## Abstract

### Background

Reporting two million human Lassa fever (LF) cases with around 10,000 associated annual mortality, the West African sub-region is endemic for Lassa fever virus (LASV). The true incidence of LF is difficult to determine because most LASV-infected individuals show no differentiating clinical signs and symptoms. We investigated the distribution of cases, post-hospitalization survival patterns, and evaluated factors contributing to infection and clinical course of the disease during an outbreak of LF in Ondo State, Nigeria, from 2017 to 2021.

### Methods

We extracted LF data from the Integrated Disease Surveillance and Response weekly report of the Nigerian Centre for Disease Control for 2017–2021. Kaplan-Meier estimate was used to describe the probability of survival among the LF cases. Also, a univariable binary logistic regression was used to explore factors associated with mortality among the study participants. Key informant was interviewed and environmental assessments were also done.

**Data availability statement:** All relevant data are within the paper and its Supporting Information files

**Funding:** The author(s) received no specific funding for this work.

**Competing interests:** The authors have declared that no competing interests exist.

## Results

LASV infection was confirmed in 1,115 (24.5%) of 4,551 cases with clinical signs suggestive of LF (age 35.24±20.77) and case fatality rate of 25.5%. Hospitalized patients who did not recover within 17 days had less than 50% chance of survival. Age is a strong predictor of survival; hospitalized patients >40 years were significantly more likely than younger ones to experience mortality (Odds ratio:2.46; 95% CI=1.67–3.62; p<0.01). Similarly, male patients were significantly less likely than the females to survive beyond 10 days of hospitalization. Open sun drying of food items and congested urban residential settings with history of frequent rat sightings are possible factors for the increase of LF cases in the study area.

## Conclusion

Current case definition in Ondo State identified close to 25% of laboratory confirmed LASV infection. Human activities during the dry season (October–March) are associated with increased LF cases. We propose a One Health disease surveillance approach that synchronizes farming activities with educational campaigns as a mitigation strategy against LASV infection and mortality in Nigeria.

## Introduction

Lassa fever (LF) is a neglected viral zoonosis endemic in Nigeria, Benin Republic, and the Mano River Union countries (Sierra Leone, Guinea, Liberia) of West Africa [1–4]. Sporadic outbreaks of LF have also been reported in Ghana [5], Mali, Senegal, and Cote d'Ivoire [6]. In this region, it is estimated that approximately 2 million human infections and 5,000–10,000 deaths are associated with LF annually [7]. In the healthcare environment, the disease is transmitted from person to person through contact with infected individuals, as a result of poor biosafety practices [7]. Conversely, in the general population, disease transmission is indirect and associated with contamination of food with feces and urine from the reproductively prolific multimmamate rat, *Mastomysis natalensis* [8]. Thus, in endemic countries like Nigeria, LF outbreaks are common in rural areas where agricultural practices and sociodemographic factors favor human-rodent interactions.

LF is caused by Lassa fever virus (LASV), a single-stranded, segmented, RNA virus belonging to the Arenaviridae family [1,8]. Phylogeographic analyses have classified LASV into six main clades that appear to be associated with specific geographic areas in West Africa. Isolates belonging to clades I – III are found in Nigeria; clade IV exists in Sierra Leone, Guinea, and Liberia; clade V is described in southern Mali, while recent isolates from Togo were classified as clade VI. Although the seeming geographic restriction of LASV genotypes suggests adaptation to local reservoir hosts, like most RNA viruses, continuous evolution of isolates within their natural nests will ensure the emergence of new phenotypic and genotypic variants over time [9,10]. Such changes may result in different clinical presentation of LF and affect the

course and outcomes of infection. It is therefore important to evaluate changes in the course of disease in endemic areas to identify patterns that may reflect altered pathogenicity of local genetic variants of LASV.

Lassa fever occurs in humans of all age group and both sexes. Humans get infected with LF virus (LFV) mainly via contact with urine or faeces of infected *Mastomys* rats, other rodents, or blood and other bodily secretions of a person infected with Lassa fever [11–13]. Infection can also occur within community and health-care settings, where virus may be spread through contaminated medical equipment. Recently, sexual transmission of LFV have been reported [14,15]. The transmission of Lassa virus through animal contact highlights the need for "One Health," understanding the relationship between human living conditions and the rodent reservoir, in human disease control. In endemic areas, the disease has a specific seasonal pattern, with a noticeable increase in occurrences from November to April [15]. This period falls within the dry season, in most West Africa settings, when rats are more likely to enter human homes in quest of sustenance, thus increasing human exposure to the virus (15). Individuals living in rural, low-resource areas with poor sanitation and overcrowded living conditions are at a much higher risk of developing Lassa fever (15). Furthermore, rural communities that rely on agriculture and animals increase the possibility of rodent encounter, raising the risk of viral infection. Unfortunately, Lassa virus endemic communities with limited healthcare resources may lack adequate public health infrastructure to avoid community-wide Lassa fever outbreaks.

The incubation period of LF has been estimated as 3–21 days following LASV infection. However, up to 80% of individuals infected with LASV are asymptomatic or show mild non-differentiating clinical signs and symptoms that may not require treatment. Therefore, the disease may remain undetected in a community for a long time until severe cases occur, making it difficult to determine the true incidence of LF in an endemic population [9,16–18]. In this report, we analyzed mortality and survival patterns during a local outbreak of LF in Ondo State, Nigeria between 2017 and 2021. We evaluated sociodemographic factors associated with LASV infection and survival during an outbreak.

## Materials and methods

### Study context

This is a cross-sectional study. It involved the use of secondary data extracted from the Integrated Disease Surveillance and Response (IDSR) weekly epidemiological data line list for 2017–2021, obtained from the Epidemiology Unit of Ondo State Ministry of Health, Nigeria. Between January 1, 2017 and December 31, 2021, a total of 8,201 suspected cases of LF were reported in Nigeria with 1,067 (13.0%) confirmed cases and 189 deaths (CFR = 17.7%). Overall, 68 Local Government Areas (LGAs) across 17 states of Nigeria reported at least one confirmed case over this period in 2021. Nonetheless, three states accounted for about 84% of the cases reported (Edo: 42%; Ondo: 34% and Bauchi: 8%) [19]. Thus, Ondo State, for the first time since 2017, has now overtaken Edo State as the epicenter of LF in Nigeria [19]. This sudden upsurge in the number of LF cases in Ondo State presents an opportunity to examine demographic factors associated with mortality during the current outbreak of LF in an endemic area. In addition, we explored LF surveillance data for Ondo State from January 2017 to December 2021 with the aim of gaining better insight into the trends and possible predictors of the course of the disease in the state.

### Geographic location

Ondo State is one of the 36 states of the Federal Republic of Nigeria. It is situated in the southwestern region of the country. The state lies between longitudes 4"30" and 6" East, and latitudes 5"45" to 8" 15" North. Ondo State has a total land area of 14,788,723sq km and an estimated population of 4,883,793 (male = 2,462,525; female = 2,421,267). It is divided into 18 LGAs. The state has a typical tropical climate with two main seasons: the rainy season (April to September) and the dry season (October to March). The annual rainfall is estimated at 1,150−2,000 mm. Throughout the year, temperature fluctuates between 21°C and 29°C, and humidity is very high. Agricultural practices and other economic activities are

carried out around the seasons: planting is done during the rainy season, while harvests are dried and stored during the dry season. The State has some strategic local markets which serve as the major transaction hubs for most of the rural population.

## Data source/quality

Secondary data extracted from the Integrated Disease Surveillance and Response (IDSR) weekly epidemiological data line list from January 2017 to December 2021 were obtained from the Epidemiology Unit of Ondo State Ministry of Health, Nigeria. Only cases that met the Nigeria Centre for Disease Control (NCDC) case definition for LF and had laboratory results entered were considered for the study. Cases with pending laboratory results were excluded. Curated data were exported and saved as excel files. The current study made use of data of all patients that were available. In the dataset, the LF diagnostic result of each patient was reported as positive or negative. Patients whose results were unknown at the time of data collection were reported as missing. Continuous variables were summarized using mean+SD. For the purpose of uniformity, the age of respondents that were reported in months were converted to years.

During an in-depth interview, the State DSNO provided insights regarding the risk practices and activities driving LF outbreaks. Through this, insights on the vulnerabilities based on the observation of the environment around the neighborhoods and major market settings were also provided.

The study primarily focused on demographic factors associated with mortality and did not explore other factors such as socioeconomic status, access to healthcare.

## Definition of study variables

Independent variables: The major independent variables selected were LGA, date of onset of symptoms, date seen at the health facility, in-patient status of participants (Yes/No), length of hospital stay, sex, and age of patient. Dependent/outcome variables: The outcome variables include result of laboratory confirmatory testing for LASV infection (positive/negative), outcome (alive, dead), and year and month of disease occurrence.

## IDSR Lassa fever case definitions

**A suspected case.** Any person with an illness of gradual onset with one or more of the following: malaise, fever, headache, sore throat, cough, nausea, vomiting, diarrhea, myalgia (muscle pain), central chest pain or retrosternal pain, hearing loss, along with:

(a) either a history of contact with excreta or urine of rodents, or

(b) history of contact with a probable or confirmed LF case within a period of 21 days of onset of symptoms, or

(c) contact with any person with inexplicable bleeding.

**A probable case.** Any suspected case as defined above but who died or was lost to follow-up before samples could be collected for laboratory testing.

**A confirmed case.** Any suspected case with laboratory confirmation of recent LASV infection (positive IgM antibody, PCR, or virus isolation).

## Ethical considerations

All data were fully anonymized before they were accessed. Approval was received from the University of Ibadan/University College Hospital Institutional Review Board (UI/UCH/22/0305). This study involved secondary data analysis and there

was no contact with the patients. Access was granted to the secondary data after all forms of identifiers that could reveal the patient's identity were removed. Appropriate steps were taking to ensure confidentiality and privacy as stated in the Helsinki declaration.

### Data analysis

Data for the period under review (January 2017 to December 2021) were sorted, cleaned and relevant variables were extracted using Microsoft Excel. Cleaned data were imported into EpiInfo Software version 7.0. Descriptive statistics (mean, SD, frequency, and proportions) was used to appropriately describe the dataset. ArcGIS (Esri, Redlands, CA) was used to fine tune the geographical clustering of LF in Ondo State. Statistical analysis was carried out using Stata version 16 (StataCorp LLC, College Station, TX). The Kaplan-Meier curve was used to describe the survival pattern of LF patients. A binary logistic regression model was employed to investigate the factors associated with mortality among the LF cases. This outcome was dichotomized as coded as zero (Alive) and one (Dead). A univariable logistic regression model was then fitted to examine factors that were independently associated with mortality among the participants. The level of statistical significance ($\alpha$) was set at 5%.

Furthermore, key stakeholders including community leaders/influencers, market association/women leaders, environmental health workers and community animal health workers were interviewed (via verbal discussion). In addition, environmental assessment (via virtual observation) was carried out to gain insight into possible socio-ecological factors that could be fueling the endemicity and outbreak of LF in Owo (the epicenter) and the adjoining towns.

## Results

### Socio-demographic pattern of LF patients in Ondo State (January 2017-December 2021)

The present study involved a comprehensive analysis of a dataset of LF patients over a five-year period, from January 2017 to December 2021 (Table 1). A total of 1,115 (24.5%) out of 4,551 (Age = 35.24 ± 20.77 years) suspected cases tested positive for LF infection during the study period. Throughout this period, the majority (approximately 60%) of the suspected cases lived in Owo LGA.

### Distribution of suspected and confirmed cases, and mortality among LF cases, Ondo State, 2017–2021

A total of 12 (0.3%) cases were deemed probable because of a history of illness and death following contact with infected clinically sick and hospitalized individuals. However, they were not tested for LASV. Of 1,115 confirmed LF cases, 284 died (case fatality rate, CFR, 25.5%). The highest number of cases were reported in Owo (60.8%) and Akure South (13.7%), two LGAs that are in close proximity with each other (Figs 1 and 2). The frequency and geographic distribution of confirmed cases and mortality as well as seasonal distribution of suspected cases are summarized in Table 2 and Fig 3. Majority of the confirmed cases and deaths were reported during the first (623 confirmed, 144 deaths) and fourth (209 confirmed, 47 deaths) quarters of the year. However, most deaths occurred between April and June, the second quarter).

The 2021 dataset of 964 infected individuals was further analyzed with the primary objective of estimating the pattern of survival over time after admission to the hospital (Fig 4A). The survival curve graphically represents the probabilities of survival at different time points after admission to the hospital (Table 3). The data showed that the minimum survival period observed was 5 days post-admission, while the maximum post-admission survival period was 30 days. The chances of surviving beyond 30 days from the onset of LF are remarkably low. This implies that the disease has a high mortality rate within the first month of infection, making early diagnosis and effective treatment crucial in improving patient outcomes. These findings suggest individual variation in the course of disease in infected patients. The average time to death of LF patients was approximately 17 days following hospitalization.

**Table 1. Socio-demographic pattern of LF patients in Ondo State (2017-2021).**

| | 2017 (n = 283) n (%) | 2018 (n = 626) n (%) | 2019 (n = 1123) n (%) | 2020 (n = 1558) n (%) | 2021 (n = 964) n (%) |
|---|---|---|---|---|---|
| **Result** | | | | | |
| Negative | 207 (73.1) | 463 (74.0) | 833 (74.2) | 1132 (72.7) | 791 (82.1) |
| Positive | 76 (26.9) | 162 (25.8) | 282 (25.1) | 424 (27.2) | 171 (17.7) |
| Missing | 0 | 1 (0.2) | 8 (0.7) | 2 (0.1) | 2 (0.2) |
| **Mean age ± SD** | 37.85 ± 17.89 | 35.06 ± 20.13 | 31.82 ± 20.0 | 34.12 ± 21.29 | 37.33 ± 21.97 |
| **Age group (yrs)** | | | | | |
| < 20 | 37 (13.1) | 137 (22.1) | 306 (27.3) | 382 (24.5) | 219 (22.7) |
| 20–29 | 60 (21.2) | 122 (19.7) | 227 (20.2) | 322 (20.7) | 159 (16.5) |
| 30–39 | 60 (21.2) | 138 (22.3) | 237 (21.1) | 236 (15.2) | 161 (16.7) |
| 40–49 | 56 (19.8) | 90 (14.5) | 139 (12.4) | 200 (12.8) | 125 (13.0) |
| 50–59 | 19 (6.7) | 53 (8.6) | 82 (7.3) | 143 (9.2) | 95 (9.8) |
| 60+ | 42 (14.8) | 80 (12.9) | 115 (10.2) | 224 (14.4) | 180 (18.7) |
| Non-response | 9 (3.2) | 6 (1.0) | 17 (1.5) | 51 (3.3) | 25 (2.6) |
| **LGA** | | | | | |
| Akoko | 18 (6.4) | 32 (5.1) | 86 (7.7) | 177 (11.4) | 78 (8.1) |
| Akure | 41 (14.5) | 103 (16.5) | 189 (16.8) | 327 (21.0) | 189 (19.6) |
| Ose | 33 (11.7) | 47 (7.5) | 110 (9.8) | 101 (6.5) | 61 (6.3) |
| Owo | 182 (64.3) | 392 (62.6) | 695 (61.9) | 900 (57.8) | 600 (62.2) |
| Others | 9 (3.2) | 52 (8.3) | 43 (3.8) | 53 (3.4) | 36 (3.7) |
| **Sex** | | | | | |
| Female | 126 (44.5) | 291 (46.5) | 558 (49.8) | 794 (51.0) | 471 (49.0) |
| Male | 155 (54.8) | 332 (53.0) | 563 (50.2) | 753 (48.3) | 490 (51.0) |
| Missing | 2 (0.7) | 3 (0.5) | 2 (0.2) | 11 (0.7) | 3 (0.3) |
| **In-Patient** | | | | | |
| No | 5 (1.8) | 4 (0.6) | 1 (0.1) | 0 | 0 |
| Yes | 278 (98.2) | 621 (99.2) | 1122 (99.9) | 1558 (100.0) | 964 (100.0) |
| Unknown | 0 | 1 (0.2) | 0 | 0 | 0 |
| **Outcome** | | | | | |
| Alive | 260 (91.9) | 575 (92.4) | 1049 (93.4) | 1469 (94.3) | 913 (94.7) |
| Dead | 23 (8.1) | 47 (7.6) | 74 (6.6) | 89 (5.7) | 51 (5.3) |
| Unknown | 0 | 4 (0.6) | 0 | 0 | 0 |

In addition, we conducted a stratified survival analysis by gender (Fig 4B) to investigate potential differences in survival patterns between male and female patients. The analysis showed that male patients had a lower probability of survival during the first 10 days of hospitalization, suggesting that male patients have a higher risk of mortality in the early stages of infection. However, the analysis also indicated that male patients had relatively improved chances of survival between 10 and 20 days after admission. On the other hand, it was observed that male patients had lower survival probabilities beyond 20 days after admission, suggesting that the disease may have a more severe and prolonged impact on male patients in the later stages of infection.

Further analysis of the frequency and distribution of mortality (Table 4) revealed that out of the 171 cases that were identified in 2021, 48 patients (28.1%) died. Mortality was significantly associated with age (p = 0.012) but not the LGAs where the cases lived (p = 0.094) or gender (p = 0.388). Individuals aged 60 years and above had a higher proportion of mortality at 8.9%. Similarly, individuals in the 50–59 years age group had a mortality rate of 8.4%. These findings suggest

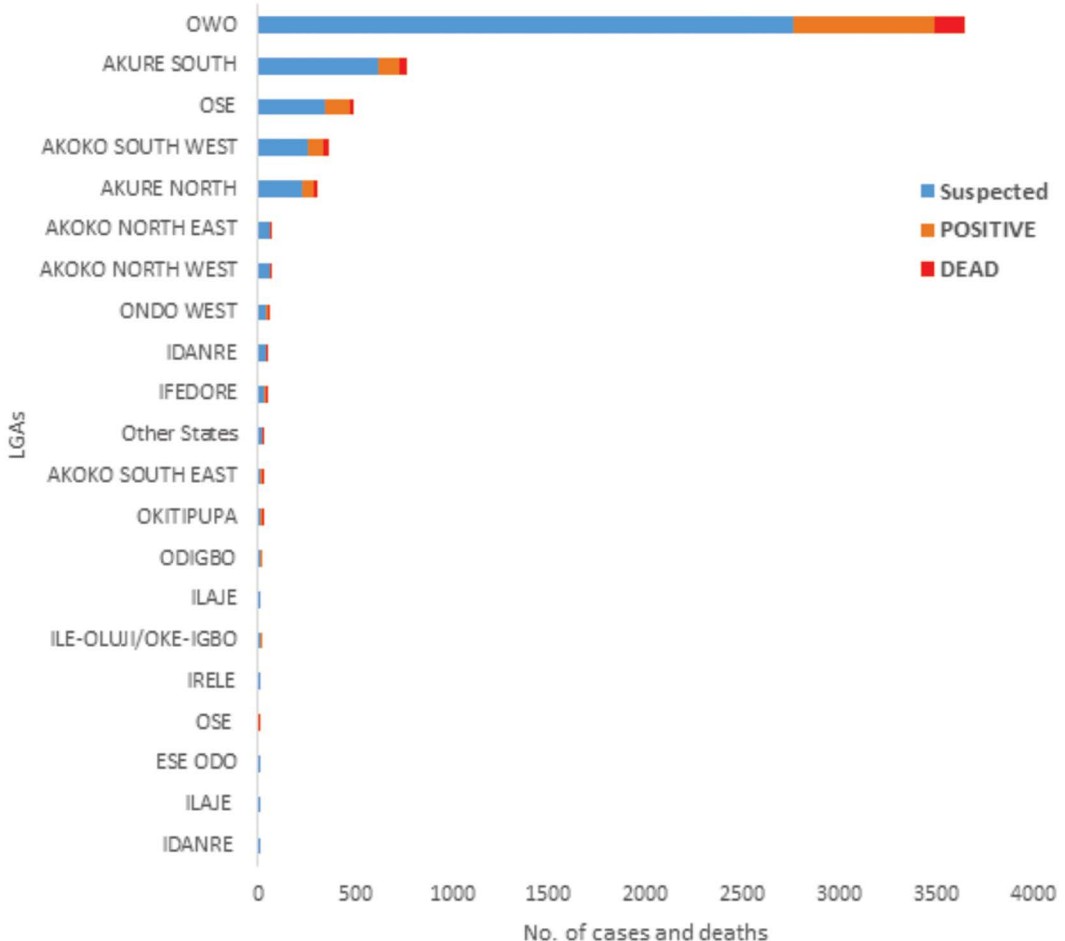

**Fig 1. The distribution of suspected and confirmed cases and deaths by LGA based on Ondo State LF data, 2017-2021.**

that older individuals with LF had a higher risk of death, potentially due to factors such as age-related immunosuppression and presence of co-morbidities.

Patients from Akure, Akoko, Ondo Central, and Ondo South were less likely to be infected with LASV compared to those from Owo and Ose LGA. The odds ratios (OR) for the respective locations were as follows: Akure (OR = 0.66; 95% CI = 0.55–0.79), Akoko (OR = 0.69; 95% CI = 0.54–0.90), Ondo Central (OR = 0.44; 95% CI = 0.28–0.72), and Ondo South (OR = 0.27; 95% CI = 0.08–0.91) (Table 5). Additionally, patients within the 20–39 years age group were 1.43 times more likely (95% CI = 1.18–1.72) to have LF compared to those below 20 years of age. Similarly, patients aged 40–59 years had a higher likelihood (OR = 1.82; 95% CI = 1.49–2.23) of being infected with LASV (Table 5).

In 2017, there were 278 reported cases of LF which resulted in 23 deaths, giving a CFR of 8.3% However, there was a consistent decrease in CFR in the subsequent years, with the rate dropping as low as 5.3% in 2021. Analysis of the CFR in different LGAs of Ondo State showed that Akoko LGA had a CFR of 8.7%, Ondo Central 8.6%, and Akure 6.2%. Furthermore, the CFR varied based on age groups with the highest value of 11.0% observed among patients aged ≥ 60 years, while the lowest CFR of 3.6% was seen among patients below 20 years of age. This highlights the increased vulnerability of older individuals to severe outcomes and complications from LF infection.

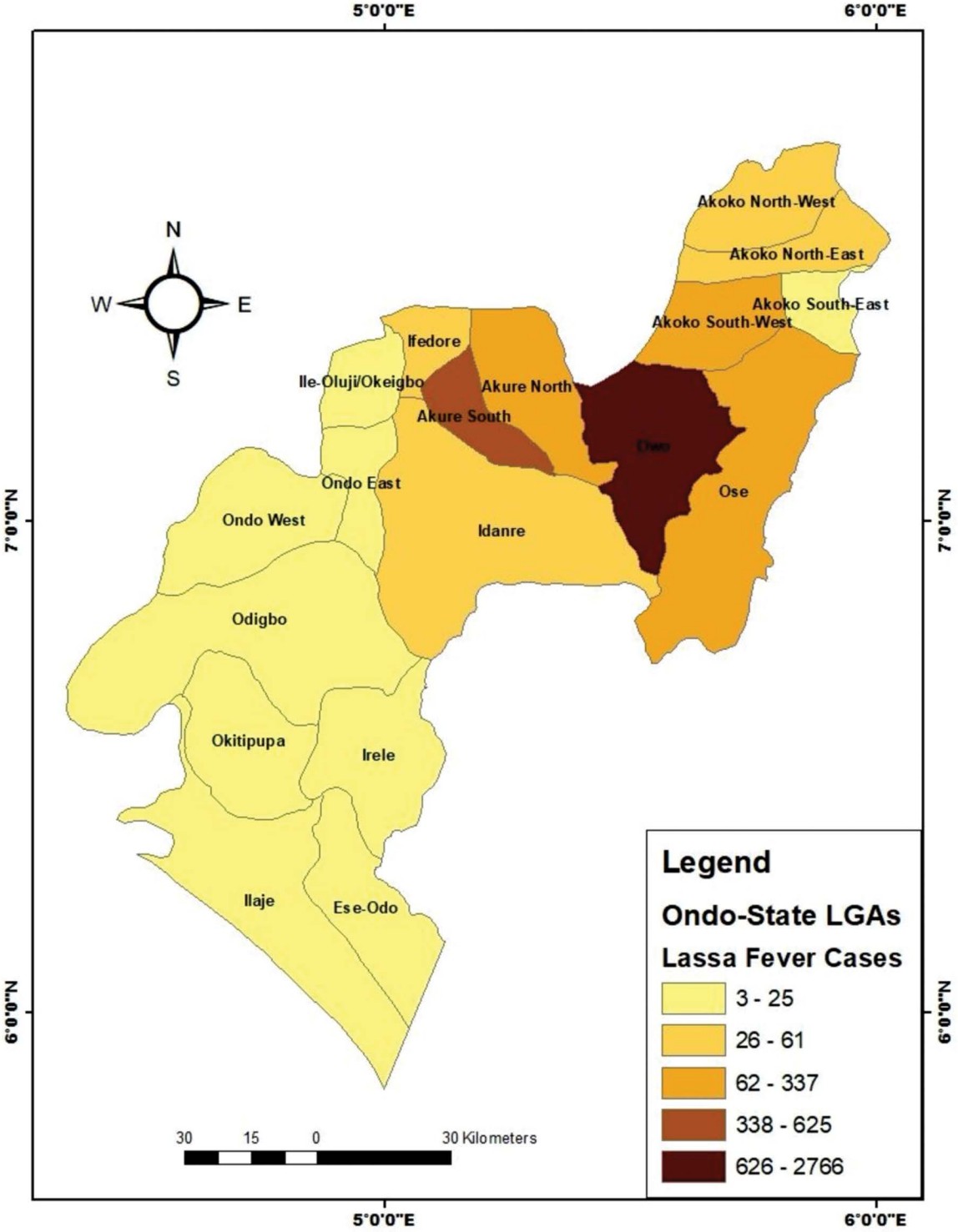

**Fig 2. Map of Ondo State showing the 18 Local Government Areas.** The intensity of the colour indicates the number of LF cases. Source: Department of Geography, University of Ibadan; https://Grid3.org. The administrative boundary used in the thematic mapping of Lassa fever across LGAs in Ondo State was obtained from the Grid 3. The link to LGA boundaries in Nigeria is: https://data.grid3.org/datasets/GRID3::grid3-nga-operational-lga-boundaries/about.

**Table 2. Distribution of suspected and confirmed cases and deaths by selected variables based on Ondo State LF data, 2017-2021.**

| Variables | Suspected | Positive | % | Dead | CFR | Probable |
|---|---|---|---|---|---|---|
| **Age group** | | | | | | |
| 0–2 yrs (Babies) | 219 | 32 | (2.9%) | 12 | (4.2%) | 2 |
| 3–16 yrs (Children) | 667 | 128 | (11.5%) | 21 | (7.4%) | 1 |
| 17–30 yrs (Young adults) | 1,237 | 302 | (27.1%) | 40 | (14.1%) | 1 |
| 31–45 yrs (Middle-aged adults) | 1,126 | 286 | (25.7%) | 73 | (25.7%) | 3 |
| > 45 yrs (Old adults) | 1,200 | 344 | (30.9%) | 133 | (46.8%) | 3 |
| Missing | 97 | 21 | (1.9%) | 5 | (1.8%) | 2 |
| **Sex** | | | | | | |
| Female | 2,236 | 530 | (47.6%) | 123 | (43.3%) | 6 |
| Male | 2,292 | 578 | (51.9%) | 161 | (56.7%) | 6 |
| Missing | 18 | 5 | (0.4%) | | | – |
| **Year** | | | | | | |
| 2017 | 278 | 74 | (6.6%) | 23 | (8.1%) | – |
| 2018 | 626 | 164 | (14.7%) | 48 | (16.9%) | – |
| 2019 | 1,121 | 280 | (25.2%) | 73 | (25.7%) | 8 |
| 2020 | 1,557 | 424 | (38.1%) | 89 | (31.3%) | 2 |
| 2021 | 964 | 171 | (15.4%) | 51 | (18.0%) | 2 |
| **Lag Time** | | | | | | |
| ≤ 7 days | 2,383 | 551 | (49.5%) | 141 | (49.6%) | 7 |
| >7 days | 2,163 | 562 | (50.5%) | 143 | (50.4%) | 5 |
| **Quarter of year** | | | | | | |
| Quarter 1 | 1,828 | 623 | (56.0%) | 144 | (50.7%) | 9 |
| Quarter 2 | 881 | 139 | (12.4%) | 52 | (18.3%) | 1 |
| Quarter 3 | 763 | 142 | (12.8%) | 41 | (14.4%) | – |
| Quarter 4 | 1,074 | 209 | (18.8%) | 47 | (16.6%) | 2 |

The results (Table 6) of regression analysis provided further insights into the factors affecting survival rates. Cases from Akoko LGA were found to be 1.54 times more likely (95% CI = 1.06–2.27) not to survive the infection. Patients aged 40–59 years were 2.46 times more likely (95% CI = 1.67–3.62) to die, while those above 60 years of age had an even higher likelihood (OR = 3.30; 95% CI = 2.22–4.92) of dying. These indicate that as age increases, the risk of mortality from LF also increases.

Key stakeholders interviews and environmental assessment identified possible socio-ecological factors that could be fueling the endemicity and outbreak of LF in Owo (the epicenter) and the adjoining towns. The major factors included sun drying of food items by the road side, the presence of local markets (Ose, Ogbese, and Elegbeda) that serve as central transaction hubs for Owo and the adjoining LGAs. Others include wastes generated from the market which attracts rats, poor and unsafe sewage disposal, proximity of refuse dump to residential areas and congested urban residential setting with rat infestation (Fig 5).

## Discussion

We carried out an epidemiological analysis of a five-year (January 2017-December 2021) LF dataset during an outbreak in Ondo State, southwestern Nigeria. The data revealed a sharp increase in the number of reported cases of LF in Ondo State from January 2017 to December 2021 and a sudden drop towards 2021. Similar observations have been reported in previous studies conducted across Nigeria [20–22]. One possible reason for this observation is mitigation efforts of the

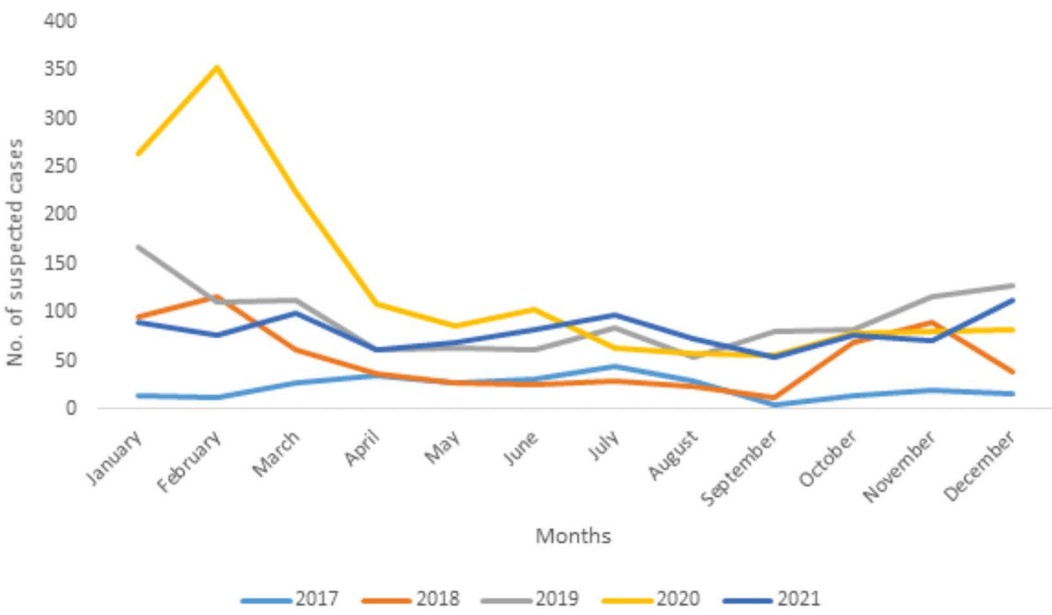

**Fig 3. Seasonal trend of suspected LF cases, Ondo State, 2017-2021.**

Nigeria Centre for Disease Control (NCDC) through increasing laboratory capacity for LF diagnosis as well as the establishment of the Surveillance Outbreak Response Management and Analysis System (SORMAS), an electronic surveillance reporting system established in 2018. These interventions have resulted in significant increase in the number of identified and reported cases of LF in the country [23]. However, the drop in the number of cases reported in 2021 might not be unconnected with the interruption occasioned by the COVID-19 pandemic of 2020. The pandemic redirected the focus of national surveillance system resources away from other public health problems. The COVID-19 pandemic was a major factor that affected the identification, treatment, and management of both infectious and non-communicable diseases globally [24].

Consistent with the findings of previous studies, our results for 2018, 2019 and 2020, showed peaks of outbreaks between the last and the first quarter of the year [22,25]. The annual peak between the last and first quarter has been attributed to a combination of factors including (i) increased *Mastomys* rodents' population, and (ii) seasonal coincidence with preparation for planting in which prevailing environmental conditions encourage human-rodent interactions [25–27]. Notably, LF has previously been associated with seasonal climatic change [23,28]. Although, the major drivers of recurring incidence and risk factors for spillover of LF are not well understood, some of the other factors that have been implicated include ineffective food storage, quality of housing, and certain agricultural practices associated with crop processing and packaging [29].

Findings from this study revealed that majority (28.7%) of the LF cases were adults aged 45 years and above. Previous studies described LF as a disease found in individuals aged 15–64 years [26,30]. Importantly, the data from this study specifically identified age as a key risk factor for LF infection, with patients 20 years and above more likely to be infected compared to those aged ≤ 19 years. This age range contains the segment of the population engaged in agricultural practices that promote exposure to LASV. Furthermore, our data showed a higher case fatality rate among individuals who are older than 40 years compared to the younger segment of the population. The increased CFR in this age group may be related to complications resulting from comorbidities. Our study did not explore co-incident health conditions in the population studied. The CFR recorded in this study is higher than the 22.5% national CRF reported by the Nigerian Centre for

A

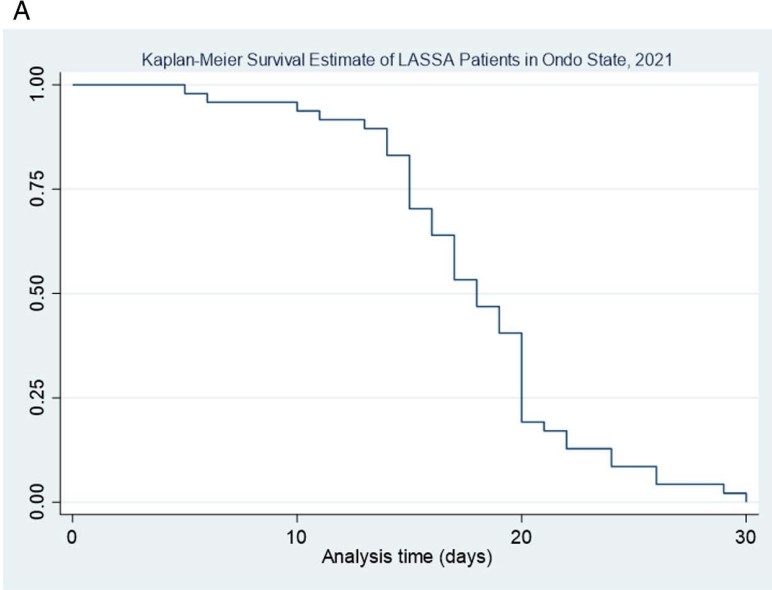

B

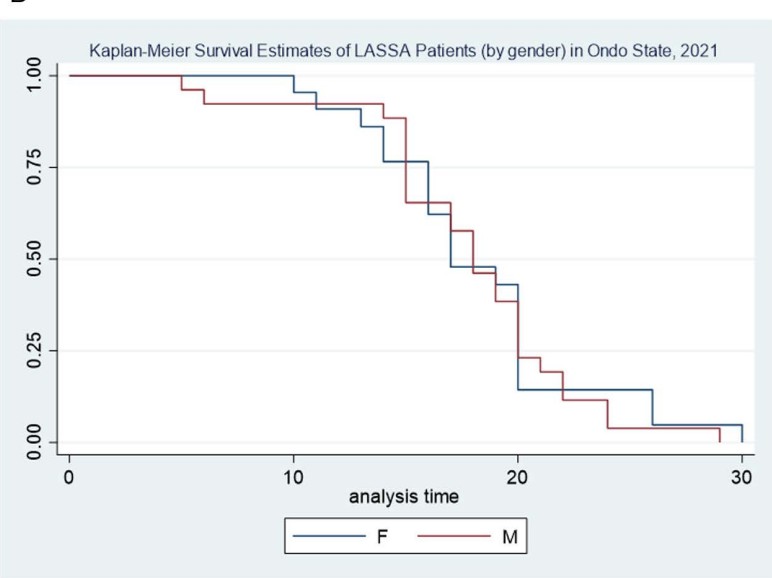

**Fig 4. A. Kaplan-Meier survival estimate of LF cases in Ondo State, 2021.** B: Kaplan-Meier survival estimate of LF cases (by gender) in Ondo State, 2021.

Disease Control within the period under consideration [19]. This higher CRF might not be unconnected with the fact that Ondo State is one of the States with the highest LF confirmed cases in Nigeria, notwithstanding the establishment of case management health facility in Irua, Edo State, a neighbouring state to Ondo State [19].

A major observation from our data is that majority (89%) of the cases occurred in Owo and adjoining LGAs (Ose, Akure North, Akure South and Akoko South) of the state. These LGAs share contiguous borders with Edo Sate, a location where LF has been known to be endemic. In addition, there are three major open-air markets – Ose, Ogbese, and Elegbeda – that link the four LGAs. These markets are the main transaction hubs for inter and intrastate commerce.

**Table 3. Probability of survival among LF patients in Ondo State (2021).**

| Length of hospitalization (days) | Probability of survival | Standard error | 95% confidence interval |
|---|---|---|---|
| 5 | 0.979 | 0.0206 | 0.861–0.997 |
| 10 | 0.937 | 0.0349 | 0.818–0.979 |
| 13 | 0.895 | 0.0443 | 0.766–0.955 |
| 17 | 0.533 | 0.0728 | 0.381–0.663 |
| 21 | 0.171 | 0.0549 | 0.079–0.289 |
| 24 | 0.085 | 0.0408 | 0.027–0.186 |
| 29 | 0.021 | 0.0211 | 0.002–0.098 |

**Table 4. Factors associated with mortality among LF patients in Ondo State, 2021.**

| | Alive | Dead | p-value |
|---|---|---|---|
| **Result** | | | |
| Negative | 790 (99.9) | 1 (0.1) | **<0.01** |
| Positive | 123 (71.9) | 48 (28.1) | |
| **Age group (yrs)** | | | |
| < 20 | 214 (97.7) | 5 (2.3) | **0.012** |
| 20–29 | 156 (98.1) | 3 (1.9) | |
| 30–39 | 151 (93.8) | 10 (6.2) | |
| 40–49 | 118 (94.4) | 7 (5.6) | |
| 50–59 | 87 (91.6) | 8 (8.4) | |
| 60+ | 164 (91.1) | 16 (8.9) | |
| **LGA** | | | |
| Akoko | 73 (93.6) | 5 (6.4) | 0.094 |
| Akure | 177 (93.7) | 12 (6.3) | |
| Ose | 54 (88.5) | 7 (11.5) | |
| Owo | 576 (96.0) | 24 (4.0) | |
| Others | 33 (91.7) | 3 (8.3) | |
| **Sex** | | | |
| Female | 449 (95.3) | 22 (4.7) | 0.388 |
| Male | 461 (94.1) | 29 (5.9) | |

Thus, they might serve as major ecological niche for interaction between humans and rodents, some of which are sold for food in these markets. Also, one of the predominant economic activities among the individuals in these communities (i.e., the LGAs) is the processing and sale of granulated cassava (garri) and cassava flour, activities that involve sun drying of food. Poor food storage, unhygienic food processing, unsafe waste disposal, and poor environmental sanitation have been strongly linked to factors that could encourage the continuous challenges posed by viral diseases like LF [31–35].

Congested urban residential settings that encourage continuous breeding of *Mastomys natalensis* rats can promote human contact and spillover LASV infection from rodents to humans [36].

The major limitation of this study is the use of secondary data which may have limited the scope of interpretation of results. Again, secondary data might not provide complete insight, and all the relevant information required to specifically answer some of the critical research questions on the epidemiology of LF in the study area. However, in a data-scarce environment like Nigeria, the use of secondary data was most valuable. Moreover, our findings provided significant insight

**Table 5. Association between LF and variables tested.**

| Variable | Category | Cases/Infection status | | OR (95%CI) | P-value |
|---|---|---|---|---|---|
| | | Positive | Negative | | |
| **Year** | 2017 | 74 (26.6) | 204 (73.4) | 1 | |
| | 2018 | 162 (25.9) | 463 (74.1) | 0.96 (0.70–1.33) | 0.89 |
| | 2019 | 290 (25.8) | 833 (74.2) | 0.95 (0.71–1.29) | 0.84 |
| | 2020 | 426 (27.3) | 1132 (72.7) | 1.04 (0.78–1.38) | 0.86 |
| | 2021 | 173 (18.0) | 791 (82.0) | 0.60 (0.44–0.82) | 0.00 |
| **LGA** | Owo & Ose | 846 (27.3) | 2257 (72.7) | 1 | |
| | Akure | 168 (19.8) | 681 (80.2) | 0.66 (0.55–0.79) | 0.00 |
| | Akoko | 81 (20.7) | 310 (79.3) | 0.69 (0.54–0.90) | 0.01 |
| | Ondo Central | 20 (14.3) | 120 (85.7) | 0.44 (0.28–0.72) | 0.00 |
| | Ondo South | 3 (9.4) | 29 (90.6) | 0.27 (0.08–0.91) | 0.04 |
| | Others | 7 (21.2) | 26 (78.8) | 0.72 (0.31–1.66) | 0.56 |
| **Age** | 0–19 | 203 (18.8) | 878 (81.2) | 1 | |
| | 20–39 | 432 (24.8) | 1311 (75.2) | 1.43 (1.18–1.72) | 0.00 |
| | 40–59 | 306 (29.6) | 727 (70.4) | 1.82 (1.49–2.23) | 0.00 |
| | ≥ 60 | 184 (26.6) | 507 (73.4) | 1.57 (1.25–1.97) | 0.00 |
| **Sex** | Male | 588 (25.5) | 1714 (74.5) | 1 | |
| | Female | 537 (23.9) | 1709 (76.1) | 1 (0.87–1.15) | 0.97 |

**Table 6. Association between outcome of infection with LASV and variables tested.**

| Variable | Category | Infection | Outcome | OR (95% CI) | P-value |
|---|---|---|---|---|---|
| | | Dead n (%) | Alive n (%) | | |
| **Year** | 2017 | 23 (8.3) | 255 (91.7) | 1 | |
| | 2018 | 47 (7.5) | 578 (92.5) | 0.90 (0.53–1.52) | 0.79 |
| | 2019 | 74 (6.6) | 1049 (93.4) | 0.78 (0.48–1.27) | 0.39 |
| | 2020 | 89 (5.7) | 1469 (94.3) | 0.67 (0.42–1.08) | 0.13 |
| | 2021 | 51 (5.3) | 913 (94.7) | 0.62 (0.37–1.03) | 0.08 |
| **LGA** | Owo & Ose | 180 (5.8) | 2923 (94.2) | 1 | |
| | Akure | 53 (6.2) | 796 (93.8) | 1.08 (0.79–1.48) | 0.68 |
| | Akoko | 34 (8.7) | 357 (91.3) | 1.54 (1.06–2.27) | 0.03 |
| | Ondo Central | 12 (8.6) | 128 (91.4) | 1.52 (0.82–2.80) | 0.24 |
| | Ondo South | 1 (3.1) | 31 (96.9) | 0.52 (0.07–3.85) | 0.79 |
| | Outside the State | 4 (12.1) | 29 (87.9) | 2.24 (0.78–6.44) | 0.25 |
| **Age** | 0-19 | 39 (3.6) | 1042 (96.4) | | |
| | 20-39 | 82 (4.7) | 1661 (95.3) | 1.32 (0.89–1.95) | 0.19 |
| | 40-59 | 87 (8.4) | 946 (91.6) | 2.46 (1.67–3.62) | 0.00 |
| | ≥ 60 | 76 (11.0) | 615 (89.0) | 3.30 (2.22–4.92) | 0.00 |
| **Gender** | Male | 161 (7.0) | 2141 (93.0) | 1 | |
| | Female | 123 (5.5) | 2123 (94.5) | 0.97 (0.69–1.37) | 0.93 |
| **Lag Period** | 1-7 day | 141 (6.5) | 2036 (93.5) | 1 | |
| | 8-14 days | 87 (6.1) | 1339 (93.9) | 0.94 (0.71–1.24) | 0.70 |
| | ≥ 15 days | 56 (5.9) | 889 (94.1) | 0.90 (0.66–1.25) | 0.62 |

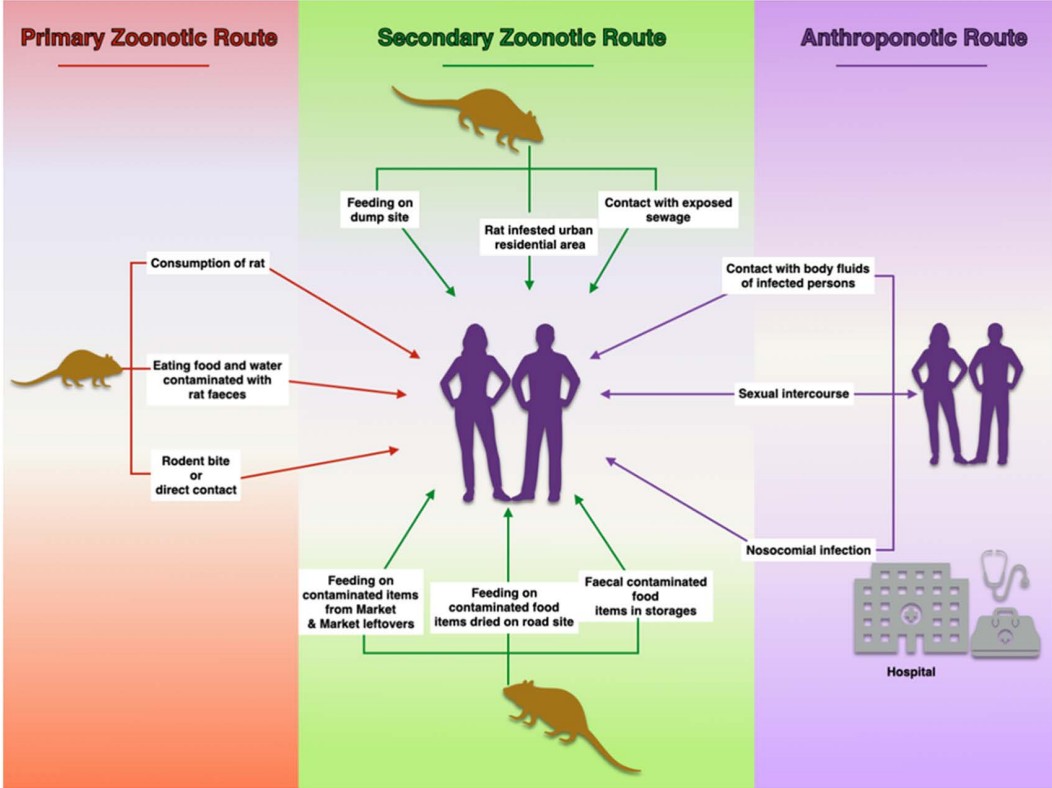

**Fig 5. Socio-ecological factors sustaining and exacerbating the endemicity and outbreak of LF in Owo and adjoining towns in Ondo State.** This Figure was generated using keynote (the presentation App on Macbook). All the images and clipart within the figure were drawn by hand with the App.

into the distribution, post-hospitalization survival patterns, and factors contributing to infection and clinical course of LF cases in the study area.

## Conclusion

Ondo State faces a significant health challenge with LF, characterized by a high case fatality rate. Mortality is predicted by age and male gender, and increased LF prevalence is associated with open sun-drying of food and living in congested urban areas with frequent rat sightings. Improved strategies for early diagnosis, patient management, and public health interventions are crucial, particularly to reduce environmental risk factors and the vulnerability of older males.

## Recommendation

To improve Lassa fever outcomes, it is essential to strengthen surveillance systems for early detection, particularly in high-risk areas. Expanding access to rapid diagnostic testing will enable timely intervention. Healthcare infrastructure must also be reinforced to ensure the availability of medications, supportive care, and trained personnel for optimal patient management. Continued research into effective treatment strategies is critical. Public health education should promote safe food handling, discourage open sun drying, and advocate for rodent-proof storage, especially among vulnerable groups such as older males and urban residents. Additionally, further studies are needed to understand the factors contributing to disease severity and the elevated mortality risk in older adults and males.

## Supporting information

**S1 File. Raw data underlining study findings.**
(XLSX)

## Acknowledgments

We acknowledge the support of the West Africa One-Health Consortium supported by the International Development Research Centre (IDRC), Canada.

## Author contributions

**Conceptualization:** Simeon Cadmus.

**Data curation:** Simeon Cadmus, Gboyega Famokun, Stephen Fagbemi.

**Formal analysis:** Victor Akinseye, Gabriel Ogunde.

**Methodology:** Victor Akinseye.

**Project administration:** Simeon Cadmus.

**Software:** Victor Akinseye.

**Supervision:** Simeon Cadmus.

**Visualization:** Victor Akinseye, Adekunle Ayinmode.

**Writing – original draft:** Victor Akinseye, Gboyega Famokun, Stephen Fagbemi, Gabriel Ogunde.

**Writing – review & editing:** Simeon Cadmus, Victor Akinseye, Eniola Cadmus, Gabriel Ogunde, Ayuba Philip, Rashid Ansumana, Adekunle Ayinmode, Olalekan Taiwo, Daniel Oluwayelu, Oyewale Tomori, Solomon Odemuyiwa.

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
