## [Decision Letter · Decision Letter 0]

Dear Dr. Cadmus,

Thank you for submitting your manuscript to PLOS ONE. After careful consideration, we feel that it has merit but does not fully meet PLOS ONE’s publication criteria as it currently stands. Therefore, we invite you to submit a revised version of the manuscript that addresses the points raised during the review process.

We look forward to receiving your revised manuscript.

Kind regards,

Muhammad Ahmad

Academic Editor

PLOS ONE

Journal Requirements:

2. In this instance it seems there may be acceptable restrictions in place that prevent the public sharing of your minimal data. However, in line with our goal of ensuring long-term data availability to all interested researchers, PLOS’ Data Policy states that authors cannot be the sole named individuals responsible for ensuring data access (http://journals.plos.org/plosone/s/data-availability#loc-acceptable-data-sharing-methods).

3. We note that [Figure 2] in your submission contain [map/satellite] images which may be copyrighted. All PLOS content is published under the Creative Commons Attribution License (CC BY 4.0), which means that the manuscript, images, and Supporting Information files will be freely available online, and any third party is permitted to access, download, copy, distribute, and use these materials in any way, even commercially, with proper attribution. For these reasons, we cannot publish previously copyrighted maps or satellite images created using proprietary data, such as Google software (Google Maps, Street View, and Earth). For more information, see our copyright guidelines: http://journals.plos.org/plosone/s/licenses-and-copyright.

Additional Editor Comments:

Revise Manuscript

Reviewers' comments:

Reviewer's Responses to Questions

**Comments to the Author**

1. Is the manuscript technically sound, and do the data support the conclusions?

Reviewer #1: Yes

Reviewer #2: Yes

Reviewer #3: Yes

2. Has the statistical analysis been performed appropriately and rigorously?

Reviewer #1: Yes

Reviewer #2: I Don't Know

Reviewer #3: Yes

3. Have the authors made all data underlying the findings in their manuscript fully available?

Reviewer #1: Yes

Reviewer #2: No

Reviewer #3: Yes

4. Is the manuscript presented in an intelligible fashion and written in standard English?

Reviewer #1: Yes

Reviewer #2: Yes

Reviewer #3: No

Reviewer #1: The authors make a strong case for using pre-existing public health data to describe the demographics of an endemic viral infection, Lassa Fever, as well as the role of poor biosafety practices in conjunction with a known rodent vector, resulting in a high disease burden. The authors’ analysis of state-level data provided by local government areas in Nigeria clearly depicts the high disease burden incurred by local populations, its significant (~25%) case-fatality rate, and highlights those patients at greatest risk.

A few suggested edits:

(lines 365-66) “Our findings showed that confirmed LF cases were mostly (28.7%) adults aged 45 years old and above.”

This conclusion is confusing and does not appear to be supported by the included data from Table 2. Per this table, ~50% of confirmed cases are adults under 45. While case-fatality rates show the greatest mortality in patients >40 with confirmed LF, the substantial majority of confirmed LF patients are <45. This point should be clarified.

(lines 372-73) “Our study did not explore co-incident health conditions in the population studied.”

The authors suggest that increased mortality in the older cohort may stem from increased comorbid conditions, while also admitting that no analysis of these “co-incident health conditions” is included in the study. Again, although these conclusions may be intuitively correct, the authors present no data to support them.

Similarly, in the abstract, Results and Discussion sections, the authors mention open air markets. sun drying of food, and congested urban areas with significant rodent presence as possible contributing factors to the increased incidence of LF within these populations, but no evidence is provided to support this contention. The authors refer to key stakeholder interviews and environmental assessments but no references or supporting documentation are included for these resources.

Secondary data use in public health surveillance is inherently limited as specific data that might be desired by the study authors may be absent or insufficiently detailed to be of use in subsequent analyses. An additional discussion of these limitations may be beneficial.

(lines 401-402) “integrate agricultural activities with health education to mitigate LF at local levels”

This leading conclusion from the study appears to be a recommendation that embraces a common-sense approach to infectious disease mitigation, although again, the rationale underlying this policy change is not supported by evidence.

Discussion of One Health strategies for LF mitigation and control may benefit from a sentence describing the WHO’s One Health concept and its role in implementing local public health solutions.

While improved biosafety practices in this limited-resource environment appears to be a logical long-term goal, the question that arises is this: Is it possible to minimize human-rodent interactions in this socioeconomic sector? While the authors point out that their study emphasized demographic factors and “did not explore other factors such as socioeconomic status, access to healthcare, and environmental conditions,” the solutions offered in the Conclusion section primarily focus on these latter factors.

In summary, the authors provide a clear picture of the demographics of patients with confirmed Lassa Fever infection in specific areas of southwestern Nigeria. Their conclusions on those most at risk for increased LF mortality are well-supported by their secondary data analysis, with a few exceptions noted above. Their public health recommendations, present throughout the manuscript and most notably in the conclusions are reasonable and likely evidence-based, however, the absence of this evidence may undermine the strength of their conclusions. These suggestions may also distract from the primary purpose of the study, a clear, concise description of the demographics of LF infection in a region where LF is an ongoing public health risk.

Reviewer #2: 1) Make the title concise like "Drivers of Lassa Fever in Southwestern Nigeria (2017-2021): An Epidemiological Study."

2) In abstract clarify the methodologies used, particularly the type of regression analysis, to provide a clearer picture to the reader. Additionally, include specific statistical results and p-values in to strengthen the abstract

3) Expand introduction section by adding brief details of Lassa virus transmission and environmental concerns and elaborate One health strategies to minimize case occurrence using suggested articles:

http://pvj.com.pk/pdf-files/24-182.pdf

http://www.pvj.com.pk/pdf-files/23-554.pdf

https://scholar.google.com/citations?view_op=view_citation&hl=en&user=RW86akAAAAAJ&citation_for_view=RW86akAAAAAJ:qxL8FJ1GzNcC

https://www.researchgate.net/profile/Ghulam-Murtaza-77/publication/378403386_Vector-Borne_Zoonotic_Diseases/links/65d88d0aadf2362b6352da06/Vector-Borne-Zoonotic-Diseases.pdf

4) Introduction could be more focused on identifying the research gap and justifying the study.

5) Discuss how the geographic features of Ondo State might influence the spread of Lassa fever to make the epidemiological significance clearer in Geographic Location

6) Address any limitations or potential biases in the data collection process, such as missing data or inconsistencies in reporting, and how they were managed in data source/quality.

7) Data analysis section needs a detailed explanation, provide more detail on the regression models, including the rationale for their selection and any assumptions made. Clarify if any corrections for multiple comparisons were applied.

8) In discussion add more referral studies; compare your findings with similar studies from Nigeria or other West African countries. Highlight any novel insights or contradictions with existing research.

9) In the discussion 4th paragraph, Lines 374-385 should be rewritten as data is not well organized.

10) The conclusion looks too general; focus on key findings of this study and mention future interventions that can be done.

11) Figures are too blurry to understand, improve quality, cite figures in running text.

12) Label Graph Y axis, what is it indicating?

Reviewer #3: 1) Lack of coherence in data presentation

2) Rewrite your abstract for further clarification

3) Significance ( P value) is not mentioned in the abstract

4) Modify the images and put the references in the given text as well.

5) See it again Source: Department of Geography, University of Ibadan; https://Grid3.org

6) https://www.ijvets.com/pdf-files/24-449.pdf

7) One health initiative ……..http://www.pvj.com.pk/pdf-files/24-035.pdf

8) https://www.ijvets.com/pdf-files/23-328.pdf

9) https://journals.lww.com/international-journal-of-surgery/fulltext/2023/04000/Re_emergence_of_the_Lassa_virus_in_Africa__a.45.aspx

Suggested for reading, understanding and supporting the manuscript

**Do you want your identity to be public for this peer review?** For information about this choice, including consent withdrawal, please see our Privacy Policy

Reviewer #1: No

Reviewer #2: No

Reviewer #3: No

---

## [Author Response · Author response to Decision Letter 1]

12 Nov 2024

The authors appreciate the editor for the oppurtunity to reviwed the manuscript again. Below are the responses to all the comments made by the reviewer:

Reviewer #1: The authors make a strong case for using pre-existing public health data to describe the demographics of an endemic viral infection, Lassa Fever, as well as the role of poor biosafety practices in conjunction with a known rodent vector, resulting in a high disease burden. The authors’ analysis of state-level data provided by local government areas in Nigeria clearly depicts the high disease burden incurred by local populations, its significant (~25%) case-fatality rate, and highlights those patients at greatest risk.

A few suggested edits:

(lines 365-66) “Our findings showed that confirmed LF cases were mostly (28.7%) adults aged 45 years old and above.”

Response: The sentence has been edited as:

“Findings from this study revealed that majority (28.7%) of the LF cases were adults aged 45 years and above”. Check lines 364-365.

This conclusion is confusing and does not appear to be supported by the included data from Table 2. Per this table, ~50% of confirmed cases are adults under 45. While case-fatality rates show the greatest mortality in patients >40 with confirmed LF, the substantial majority of confirmed LF patients are <45. This point should be clarified.

Response: Table 2 has been thoroughly reviewed, to capture the statement in the conclusion.

(lines 372-73) “Our study did not explore co-incident health conditions in the population studied.

”The authors suggest that increased mortality in the older cohort may stem from increased comorbid conditions, while also admitting that no analysis of these “co-incident health conditions” is included in the study. Again, although these conclusions may be intuitively correct, the authors present no data to support them.

Response: Thank you for this comment. The authors are not making any categorical assertion here. As the reviewer rightly pointed out, underlining health conditions is a common occurrence as individuals get older. It is against this backdrop that the authors associate likelihood of increased CFR to older individuals.

Similarly, in the abstract, Results and Discussion sections, the authors mention open air markets. sun drying of food, and congested urban areas with significant rodent presence as possible contributing factors to the increased incidence of LF within these populations, but no evidence is provided to support this contention. The authors refer to key stakeholder interviews and environmental assessments but no references or supporting documentation are included for these resources.

Response: The authors appreciate the reviewer for this comment The State DSNO provided information regarding possible risk practices or activities driving LF outbreaks. Additionally, other hazards and vulnerabilities noticed through the observation of the environment around the neighbourhood and major market settings were reported. None of these activities involved a standard questionnaire administration or checklist assessment, thus there is no formal document to present. However, a statement has been added for better clarity. Check line 226/ 227.

Secondary data use in public health surveillance is inherently limited as specific data that might be desired by the study authors may be absent or insufficiently detailed to be of use in subsequent analyses. An additional discussion of these limitations may be beneficial.

Response: Additional statement has been included. Check line 395-397

(lines 401-402) “integrate agricultural activities with health education to mitigate LF at local levels”

This leading conclusion from the study appears to be a recommendation that embraces a common-sense approach to infectious disease mitigation, although again, the rationale underlying this policy change is not supported by evidence.

Response: The authors agree with the reviewer. This conclusion was inferred from the findings of the study, and it was intended to serve as a recommendation that will help in the mitigation of the present challenge of LF.

Discussion of One Health strategies for LF mitigation and control may benefit from a sentence describing the WHO’s One Health concept and its role in implementing local public health solutions.

Response: Thank you. A statement on the WHO’s One Health concept, and its role in implementing local public health solutions has been added. Check lines 408-415.

While improved biosafety practices in this limited-resource environment appears to be a logical long-term goal, the question that arises is this: Is it possible to minimize human-rodent interactions in this socioeconomic sector? While the authors point out that their study emphasized demographic factors and “did not explore other factors such as socioeconomic status, access to healthcare, and environmental conditions,” the solutions offered in the Conclusion section primarily focus on these latter factors.

In summary, the authors provide a clear picture of the demographics of patients with confirmed Lassa Fever infection in specific areas of southwestern Nigeria. Their conclusions on those most at risk for increased LF mortality are well-supported by their secondary data analysis, with a few exceptions noted above. Their public health recommendations, present throughout the manuscript and most notably in the conclusions are reasonable and likely evidence-based, however, the absence of this evidence may undermine the strength of their conclusions. These suggestions may also distract from the primary purpose of the study, a clear, concise description of the demographics of LF infection in a region where LF is an ongoing public health risk.

Response: Thank you for the observation. Necessary adjustments have been made. Check line 226 & 227.

Reviewer #2: 1) Make the title concise like "Drivers of Lassa Fever in Southwestern Nigeria (2017-2021): An Epidemiological Study."

Response: The title has been changed as suggested.

2) In abstract clarify the methodologies used, particularly the type of regression analysis, to provide a clearer picture to the reader. Additionally, include specific statistical results and p-values in to strengthen the abstract.

Response: This section of the manuscript has been edited as shown below:

“Kaplan-Meier estimate was used to describe the probability of survival among the LF cases. Also, a univariable binary logistic regression was used to explore factors associated with mortality among the study participants”. Check lines 44-46.

3) Expand introduction section by adding brief details of Lassa virus transmission and environmental concerns and elaborate One health strategies to minimize case occurrence using suggested articles:

http://pvj.com.pk/pdf-files/24-182.pdf

http://www.pvj.com.pk/pdf-files/23-554.pdf

https://scholar.google.com/citations?view_op=view_citation&hl=en&user=RW86akAAAAAJ&citation_for_view=RW86akAAAAAJ:qxL8FJ1GzNcC

https://www.researchgate.net/profile/Ghulam-Murtaza-77/publication/378403386_Vector-Borne_Zoonotic_Diseases/links/65d88d0aadf2362b6352da06/Vector-Borne-Zoonotic-Diseases.pdf

Response: Thank you. Information from the suggested references has been included in the introduction section. Check lines 124-139.

4) Introduction could be more focused on identifying the research gap and justifying the study.

Response: Thank you. Suggested comments have been addressed.

5) Discuss how the geographic features of Ondo State might influence the spread of Lassa fever to make the epidemiological significance clearer in Geographic Location.

Response: Thank you. More details have been provided to capture the association of geographical features of Ondo State and the spread of LF. Check lines 169-173.

6) Address any limitations or potential biases in the data collection process, such as missing data or inconsistencies in reporting, and how they were managed in data source/quality.

Response: Relevant information has been included. Check lines 212-222.

7) Data analysis section needs a detailed explanation, provide more detail on the regression models, including the rationale for their selection and any assumptions made. Clarify if any corrections for multiple comparisons were applied.

Response: This section was revised as below:

“A binary logistic regression model was employed to investigate the factors associated with mortality among the LF cases. This outcome was dichotomized as coded as zero (Alive) and one (Dead). A univariable logistic regression model was then fitted to examine factors that were independently associated with mortality among the participants”. Check lines 218-221.

8) In discussion add more referral studies; compare your findings with similar studies from Nigeria or other West African countries. Highlight any novel insights or contradictions with existing research.

Response: Thank you. More references have been added.

9) In the discussion 4th paragraph, Lines 374-385 should be rewritten as data is not well organized.

Response: Thank you. This section has been reviewed as suggested.

10) The conclusion looks too general; focus on key findings of this study and mention future interventions that can be done.

Response: Thank you. The conclusion has been reviewed.

11) Figures are too blurry to understand, improve quality, cite figures in running text.

Response: These have been corrected, thank you.

12) Label Graph Y axis, what is it indicating?

Response: Thank you for the observation. Figures have been well labelled.

Reviewer #3: 1) Lack of coherence in data presentation

Response: Data presentation has been thoroughly reviewed.

2) Rewrite your abstract for further clarification

Response: Abstract has been reviewed for clarity.

3) Significance ( P value) is not mentioned in the abstract

Response: The statement was recast as: “Age is a strong predictor of survival; hospitalized patients >40 years were significantly more likely than younger ones to experience mortality (Odds ratio:2.46; 95% CI=1.67–3.62; p<0.01).

The above statement is to indicate that increasing age increases the likelihood of mortality among participants, also providing the p-value

4) Modify the images and put the references in the given text as well.

Response: Thank you. All suggestion has been incorporated.

5) See it again Source: Department of Geography, University of Ibadan; https://Grid3.org

Response: Thank you for this comment. The map was generated from the source provided and the link (https://Grid3.org).

---

## [Decision Letter · Decision Letter 1]

Dear Dr. Cadmus,

We look forward to receiving your revised manuscript.

Kind regards,

Muhammad Ahmad

Academic Editor

PLOS ONE

Journal Requirements:

Reviewers' comments:

Reviewer's Responses to Questions

**Comments to the Author**

Reviewer #1: All comments have been addressed

Reviewer #4: (No Response)

2. Is the manuscript technically sound, and do the data support the conclusions?

Reviewer #1: Yes

Reviewer #4: Partly

3. Has the statistical analysis been performed appropriately and rigorously?

Reviewer #1: Yes

Reviewer #4: Yes

4. Have the authors made all data underlying the findings in their manuscript fully available?

Reviewer #1: Yes

Reviewer #4: (No Response)

5. Is the manuscript presented in an intelligible fashion and written in standard English?

Reviewer #1: Yes

Reviewer #4: Yes

Reviewer #1: (No Response)

Reviewer #4: The efforts of the authors to give appropriate response to the earlier reviewers is recognised and commended. However, some areas especially the discussion needs more attention.

The result from the study reads: Of 1,115 confirmed LF cases, 284 died (case fatality rate, CFR, 25.5%).

Line 249

To enrich this discussion, can you compare the CFR in Ondo State with the national CFR? Line 378-379 This will help in assessing the successes in case management and the effectiveness of other programs aimed at mitigating the impact of Lassa fever

The conclusion of the study did not stem from the data presented in the study.

Authors are advised to work on the discussion and conclusion sections and clearly indicate the ares of recommendation

**Do you want your identity to be public for this peer review?** For information about this choice, including consent withdrawal, please see our Privacy Policy

Reviewer #1: No

Reviewer #4: **Yes: ** Dr Chukwuyem Abejegah

---

## [Author Response · Author response to Decision Letter 2]

13 May 2025

The authors wish to sincerely appreciate the editors and the reviewer for all the efforts in improving the quality of our manuscript. All the issues raised have been adequately addressed; find below the responses.

Reviewer #4: The efforts of the authors to give appropriate response to the earlier reviewers is recognised and commended. However, some areas especially the discussion needs more attention.

The result from the study reads: Of 1,115 confirmed LF cases, 284 died (case fatality rate, CFR, 25.5%).

Line 249

To enrich this discussion, can you compare the CFR in Ondo State with the national CFR? Line 378-379 This will help in assessing the successes in case management and the effectiveness of other programs aimed at mitigating the impact of Lassa fever.

Response: Thank you for the comment. The issue raised has been duly addressed to enrich the discussion. Check lines 379-384.

The conclusion of the study did not stem from the data presented in the study.

Authors are advised to work on the discussion and conclusion sections and clearly indicate the ares of recommendation.

Response: The conclusion has been reviewed as suggested. Recommendation section has been clearly indicated.

---

## [Decision Letter · Decision Letter 2]

Drivers of Lassa Fever in an Endemic Area of Southwestern Nigeria (2017-2021): An Epidemiological Study

PONE-D-24-05209R2

Dear Dr. Cadmus,

We’re pleased to inform you that your manuscript has been judged scientifically suitable for publication and will be formally accepted for publication once it meets all outstanding technical requirements.

Kind regards,

Muhammad Ahmad

Academic Editor

PLOS ONE

Additional Editor Comments (optional):

Reviewers' comments:

Reviewer's Responses to Questions

**Comments to the Author**

Reviewer #1: All comments have been addressed

Reviewer #4: All comments have been addressed

2. Is the manuscript technically sound, and do the data support the conclusions?

Reviewer #1: Yes

Reviewer #4: Yes

3. Has the statistical analysis been performed appropriately and rigorously?

Reviewer #1: Yes

Reviewer #4: Yes

4. Have the authors made all data underlying the findings in their manuscript fully available?

Reviewer #1: Yes

Reviewer #4: Yes

5. Is the manuscript presented in an intelligible fashion and written in standard English?

Reviewer #1: Yes

Reviewer #4: Yes

Reviewer #1: (No Response)

Reviewer #4: (No Response)

**Do you want your identity to be public for this peer review?** For information about this choice, including consent withdrawal, please see our Privacy Policy

Reviewer #1: No

Reviewer #4: **Yes: ** Dr Chukwuyem Abejegah

---

## [Editor Report · Acceptance letter]

PONE-D-24-05209R2

PLOS ONE

Dear Dr. Cadmus,

I'm pleased to inform you that your manuscript has been deemed suitable for publication in PLOS ONE. Congratulations! Your manuscript is now being handed over to our production team.

Kind regards,

on behalf of

Mr. Muhammad Ahmad

Academic Editor

PLOS ONE